# Squeezing formaldehyde into $C_{60}$ fullerene

**Vijyesh K. Vyas** [1], **George R. Bacanu**[1], **Murari Soundararajan** [1], **Elizabeth S. Marsden**[1], **Tanzeeha Jafari**[2,3], **Anna Shugai**[2], **Mark E. Light** [1], **Urmas Nagel** [2], **Toomas Rõõm** [2] ✉, **Malcolm H. Levitt** [1] ✉ & **Richard J. Whitby** [1] ✉

The cavity inside fullerene $C_{60}$ provides a highly symmetric and inert environment for housing atoms and small molecules. Here we report the encapsulation of formaldehyde inside $C_{60}$ by molecular surgery, yielding the supermolecular complex $CH_2O@C_{60}$, despite the 4.4 Å van der Waals length of $CH_2O$ exceeding the 3.7 Å internal diameter of $C_{60}$. The presence of $CH_2O$ significantly reduces the cage HOMO-LUMO gap. Nuclear spin-spin couplings are observed between the fullerene host and the formaldehyde guest. The rapid spin-lattice relaxation of the formaldehyde [13]C nuclei is attributed to a dominant spin-rotation mechanism. Despite being squeezed so tightly, the encapsulated formaldehyde molecules rotate freely about their long axes even at cryogenic temperatures, allowing observation of the ortho-to-para spin isomer conversion by infrared spectroscopy. The particle in a box nature of the system is demonstrated by the observation of two quantised translational modes in the cryogenic THz spectra.

On its discovery in 1985[1] it was recognised that fullerene $C_{60}$ contains an almost spherical 3.7 Å diameter cavity which is capable of encapsulating other species. The implantation of noble gas atoms into $C_{60}$ under extreme conditions and in very low yields followed[2,3]. A great advance has been the development of molecular surgery, involving the opening of the fullerene cages by chemical reactions, insertion of a small molecule or an atom into each cage, and subsequent closure by further reactions. The procedure has recently been reviewed[4]. Systems synthesised this way include $H_2@C_{60}$[5–7], $He@C_{60}$[7,8], $H_2O@C_{60}$[6,9], $Ne@C_{60}$[7], $HF@C_{60}$[10], $CH_4@C_{60}$[11], $Kr@C_{60}$[12], and $Ar@C_{60}$[13].

The encapsulation of atoms and molecules inside $C_{60}$ provides a unique environment for the encapsulated species. The fullerene cages isolate the encapsulated molecules from each other, preventing intermolecular bonding and causing the guest molecules to behave, to some extent, as if they are in the rarefied gas phase, even at cryogenic temperatures in the solid state, allowing detailed study of quantum effects[14–16]. At low temperatures, the tight confinement leads to translational quantisation, the study of which can be used to determine potential energy surfaces[17]. Ortho- and para spin isomers are observed for $H_2@C_{60}$[18,19] and $H_2O@C_{60}$[20–23]. The interconversion of the spin

isomers influences macroscopic properties such as the dielectric constant[22]. The quantised rotational and translational states of the confined molecules interact through rotation-translation coupling[24,25]. Unconventional confinement-induced spin-spin [0]J-couplings between endohedral [3]He and the cage [13]C nuclei have been observed[26]. Endohedral fullerenes are a frequent target for theoretical calculations and the provision of experimental data on them provides an important benchmark, particularly for non-covalent interactions[27–30].

Formaldehyde $CH_2O$ is often regarded as a model polyatomic system and has been subject to intense spectroscopic, theoretical, and computational study[31]. Monomeric formaldehyde has two hydrogens with nuclear spin $I = 1/2$, giving rise to two nuclear spin isomers with total nuclear spin $I = 1$ and 0, denoted ortho- and para-formaldehyde[32]. The interconversion of the two spin isomers has been studied in the gas phase by using selective UV laser photolysis to destroy one of the spin isomers leaving a non-equilibrium mixture[33–35]. The mechanism and rate of ortho-para interconversion has been investigated theoretically[36,37]. The ortho/para ratio of formaldehyde has been used to estimate the temperature in a variety of interstellar environments[38–40]. However, as far as we know, the spin-isomer

[1]School of Chemistry, University of Southampton, SO17 1BJ Southampton, UK. [2]National Institute of Chemical Physics and Biophysics, Akademia tee 23, 12618 Tallinn, Estonia. [3]Department of Cybernetics, Tallinn University of Technology, Ehitajate tee 5, 19086 Tallinn, Estonia. ✉e-mail: toomas.room@kbfi.ee; mhl@soton.ac.uk; rjw1@soton.ac.uk

conversion of formaldehyde has never been observed experimentally in the condensed phase.

C₆₀ cages provide the ideal nano-container for the study of monomeric formaldehyde molecules at low temperature. However, at first sight, formaldehyde seems to be too big to fit inside C₆₀. A formaldehyde molecule has a longest van der Waals dimension of 4.38 Å parallel to the CO bond, considerably larger than the 3.7 Å diameter internal space of C₆₀. Although formaldehyde has been successfully accommodated in the bowl-like cavities of open-cage fullerenes[41,42], the chemical closure of the open fullerene cages, with complete encapsulation of the formaldehyde molecules, has not been achieved.

We now report the successful encapsulation of formaldehyde ($CH_2O$) molecules inside closed C₆₀ cages. The compound $CH_2O@C_{60}$ displays some remarkable physical properties, including the spin-isomer conversion of the formaldehyde guest molecules in the cryogenic solid state, the spatial quantisation of the encapsulated molecules, confinement-induced internuclear couplings between the host and the guest nuclei, and the modulation of the fullerene electronic structure upon accommodation of the oversized guest molecule.

## Results & discussion
### Synthesis of $CH_2O@C_{60}$ and isotopologues

Murata and co-workers[42] inserted $CH_2O$ through the 17-membered orifice of **1** using high temperature and pressure (trioxane, 150 °C, 8000 atm, 35% incorporation) and observed that the $CH_2O$ escaped at room temperature. Reducing one of the carbonyls on the orifice to an alcohol, prevented the escape of $CH_2O$. Density Function Theory (DFT) calculations indicated that the high temperatures and pressures used to insert formaldehyde into the sulfide **1** should not be necessary (Supplementary Table 6). Calculations also showed that the activation energy for loss of $CH_2O$ from the sulfoxide **2** is 13 kJ mol⁻¹ higher than from the sulfide **1** suggesting that our previously reported[11] method of photochemical orifice contraction of **2** might be possible without substantial loss of $CH_2O$. Passing a stream of formaldehyde gas, generated by pyrolysis of paraformaldehyde[43], into a solution of sulfide **1** in THF at 50 °C gave $CH_2O@1$ with 70% incorporation of the endohedral molecule. THF was removed under vacuum at room temperature and the residue dissolved in toluene and filtered through a short column of silica to remove paraformaldehyde. Immediate oxidation by addition of a cold solution of dimethyldioxirane[11] gave the sulfoxide

$CH_2O@2$ with 70% filling, the remainder being $H_2O@2$ with <5% empty **2** (Fig. 1). The $CH_2O$ protons appeared as a singlet at $\delta = -0.95$ ppm in the ¹H NMR, and could be compared with the endohedral $H_2O$ signal at −10.5 ppm. The filling could also be assessed from one of the alkene protons in $CH_2O@2$ which was separated from the equivalent $H_2O@2$ and **2** signals (Supplementary Section 1.1). The half lives for loss of $CH_2O$ from $CH_2O@1$ were measured as 7.9 h at 40 °C and 70 h at 25 °C, whereas from $CH_2O@2$ it was 35 h at 55 °C (Supplementary Fig. 17, Tables 1 and 2).

Photochemical SO extrusion from $CH_2O@2$ to give the hemiacetal $CH_2O@3$ was achieved by stirring $CH_2O@2$ in a solvent mixture of THF / Toluene/ acetic acid (10% v/v aq.) using 3 × 100 W yellow LED lights for 18 h at 55 °C[11,12]. The use of THF instead of our previously reported $CH_3CN$ allowed higher concentrations to be used. This combination furnished a mixture of products $H_2O@3$ and $CH_2O@3$ in 36% yield. Since the hemiacetal **3** is rather unstable the mixture was reacted with triphenylphosphine to give $CH_2O@4$ and $H_2O@4$ in 82% yield with 35% filling of $CH_2O$. Allowing for the change in filling factors the overall yields for $CH_2O@4$ and $H_2O@4$ can be estimated as 15% and 64% respectively. For comparison previously reported yields over these steps of $CH_4@4$, $Ar@4$ and $Kr@4$ were 13%, 23% and 21%, although an improved procedure using a more powerful light source was used in the current method. The NMR of $CH_2O@4$ showed a singlet for endohedral $CH_2O$ at $\delta = 0.80$ ppm (cf. $H_2O@4$ at $\delta = -8.80$ ppm). On some occasions unreacted $CH_2O@2$ could be isolated during the purification of $CH_2O@3$ and was found to have an increased filling factor indicating that the drop in filling factor of $CH_2O@4$ is mostly due to inhibition of the closure steps by endohedral $CH_2O$ rather than loss of $CH_2O$ from $CH_2O@2$. Although leaving the photochemical reaction on for longer gave increased filling of $CH_2O$ in **3**, the yield dropped substantially due to further photochemical reactions of the desired product so it was less productive.

As an aside, $CO_2@2$ (81% filled) was easily prepared by exposure of solid **1** to 69 atm $CO_2$ at 100 °C for 18 h followed by DMDO oxidation[44]. Exposure of $CO_2@2$ to the photochemical orifice contraction conditions gave no $CO_2@3$ indicating that the large size of the endohedral molecule is stopping the desired reaction.

Finally, the orifice of $CH_2O@4$ was closed back to the C₆₀ cage using same conditions to those given in our earlier reports (Fig. 1)[7]. A combined sample of $CH_2O@4$ from several preparations was used

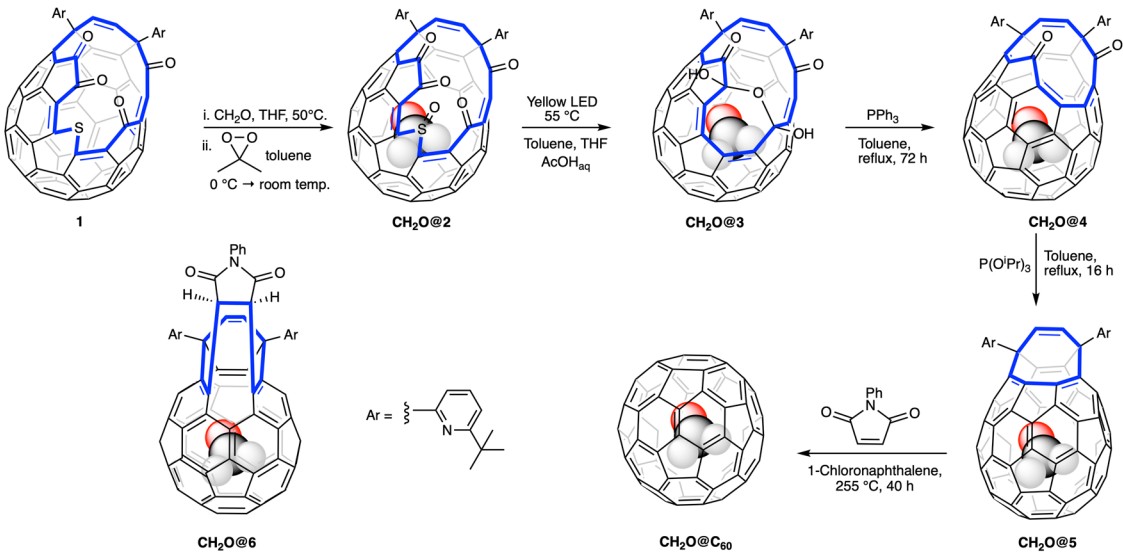

**Fig. 1 | Synthesis of $CH_2O@C_{60}$.** Sulfide **1** is 70% filled with formaldehyde ($CH_2O$) using a solution of the monomer, then oxidised to the sulfoxide **2** before photochemically induced loss of sulfur monoxide (SO) gave the orifice contracted bis-hemiacetal **3**. Phosphine and phosphite induced deoxygenative ring closures to

give $CH_2O@5$ followed by a thermal extrusion reaction gave $CH_2O@C_{60}$. The route for labelled materials $CD_2O@C_{60}$ and $^{13}CH_2O@C_{60}$ were identical except that the initial filling was carried out by heating **1** with paraformaldehyde in a sealed tube which gave only 25% incorporation of formaldehyde.

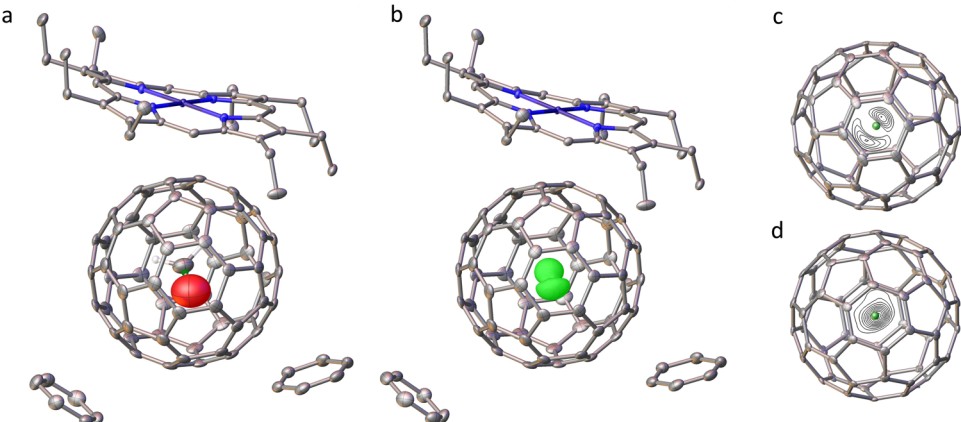

**Fig. 2 | Single crystal x-ray structure of the nickel(II) octaethylporphyrin/benzene solvate of CH₂O@C₆₀.** Recorded at 100 K with CCDC deposition number 2126579 (R1 = 0.050). **a** Thermal ellipsoids drawn at 50% probability, hydrogens, except on the CH₂O, are omitted for clarity. **b** Representation showing the observed electron density at the CH₂O location. Electron density surface drawn at the 2.1 e Å³ level. **c** and **d** Orthogonal views of difference electron density at the centre of the C₆₀ cage (contour levels drawn at approximately 0.9 e Å³ level). The centroid of the cage carbons is shown as a green sphere.

with 25% filling of CH₂O, and a mixture of C₆₀, H₂O@C₆₀ and CH₂O@C₆₀ was obtained in 46% yield over last two steps. Recirculating preparative HPLC on a Cosmosil Buckyprep® column indicated the composition to be H₂O@C₆₀ (80%), C₆₀ (4.5%) and CH₂O@C₆₀ (15.5%) and allowed pure CH₂O@C₆₀ to be isolated. It is noteworthy that the separation of CH₂O@C₆₀ from C₆₀ on the Buckyprep® HPLC column is considerably greater than that known for other atomic or molecular endofullerenes (e.g. ratio of retention times CH₂O@C₆₀: C₆₀ is 1.081compared with 1.054 for CH₄@C₆₀: C₆₀[11]). Allowing for the filling factors the yield for CH₂O@C₆₀ can be estimated as 29% (cf. 52% for the H₂O@C₆₀ + C₆₀). Since it is not possible for CH₂O to leave the cage of **4** or **5** the drop in filling factor must be due to strong inhibition of the closure steps by the endohedral molecule. We were able to isolate the known by-product **6**[45–47] which had a substantially increased CH₂O filling factor of 37%. The isolated CH₂O@C₆₀ could be further purified by sublimation (550 °C, 10⁻⁵ bar) without decomposition.

We also made the labelled species ¹³CH₂O@C₆₀ and CD₂O@C₆₀. The high cost of the starting labelled paraformaldehydes precluded the use of a large excess as in the synthesis of the unlabelled material above. Instead the labelled paraformaldehyde (100 mg) and sulfide **1** (500 mg) were sealed in a 1/4″ diameter × 10 cm long stainless steel tube and heated at 100 °C for 3 h. Work-up and DMDO oxidation as above gave ¹³CH₂O@**2** or CD₂O@**2** in high yields with ~25% filling. Photochemical orifice contraction and PPh₃-induced ring closure, as above, gave ¹³CH₂O@**4** and CD₂O@**4** with ~15% filling, the rest being H₂O@**4** (~80%) with ~5% **4**. ¹³CH₂O@**4** was closed to ¹³CH₂O@C₆₀ as above, HPLC indicating the final material was around 5% filled. Pure ¹³CH₂O@C₆₀ was isolated by recycling HPLC and displayed a doublet (J = 173 Hz) at 3.80 ppm (toluene-*d₈*) in the proton NMR and triplet (J = 173 Hz) at 197.9 ppm in the ¹³C NMR. CD₂O@C₆₀ was made by the same method and displayed a singlet at δ = 3.76 ppm (referenced to PhCH₂D at 2.09 ppm) in the deuterium NMR.

## Crystal structure of CH₂O@C₆₀

A crystal of the nickel(II) octaethylporphyrin/benzene solvate of CH₂O@C₆₀ was obtained and subjected to X-ray crystallography. Unconstrained attempts at a structure solution gave an unreasonably short C-O bond. Treating the CH₂O molecule as a rigid fragment with idealised geometry and refining its orientation in space against the observed electron density gave a satisfactory solution (R factor 4.97%) (Fig. 2a). The large thermal ellipsoid on oxygen is one indication of high mobility of the endohedral CH₂O at 100 K. To gain greater insight the electron density due to the CH₂O molecule was calculated from the difference between that observed for the entire structure in the X-ray,

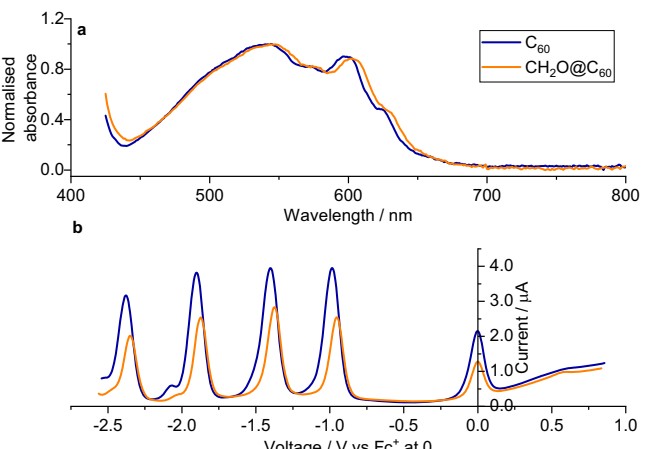

**Fig. 3 | Ultraviolet spectra and voltammetry of CH₂O@C₆₀. a** Long wavelength part of UV-vis spectra of C₆₀ and CH₂O@C₆₀ in toluene. **b** Differential Pulse Voltametry of C₆₀ and CH₂O@C₆₀ showing first four reductions relative to ferrocene (Fc/Fc⁺) at 0 V. The fullerenes were dissolved in a 4:1 mixture of toluene and acetonitrile containing 0.1 M Bu₄N.BF₄ as electrolyte. The cell contained a 3 mm diameter glassy carbon working electrode, a 1 cm² sheet of platinum as the counter electrode and a silver wire pseudo-reference electrode. Ferrocene was added as an internal standard.

and that calculated from a model containing all the atoms except the CH₂O. The residual electron density maps (Fig. 2b–d) show 2 maxima corresponding to a favoured orientation of the C-O bond axis. DFT calculations on the structure (Supplementary Fig. 27) predict the distances of the oxygen and carbon of the CH₂O molecule from the centroid of the C₆₀ (shown as a green dot in Fig. 2a, c) to be 0.86 and 0.34 Å respectively confirming the identification of these atoms in Fig. 2a. Figure 2c shows the oxygen atom distributed over a wide arc. There is no evidence that CH₂O introduces a distortion of the cage compared with the published structure of the C₆₀ complex[48] that is measurable within the resolution of the crystallographic experiment.

## UV-vis and electrochemical studies on CH₂O@C₆₀

The UV-vis absorption spectra of CH₂O@C₆₀ (Fig. 3a) showed the longest wavelength absorption peak to be red-shifted by ~6 nm from C₆₀. This shift corresponds to an energy change of −18 meV. Using a recent comparison of calculations to experimental values[49], the change in the Highest Occupied Molecular Orbital (HOMO) – Lowest

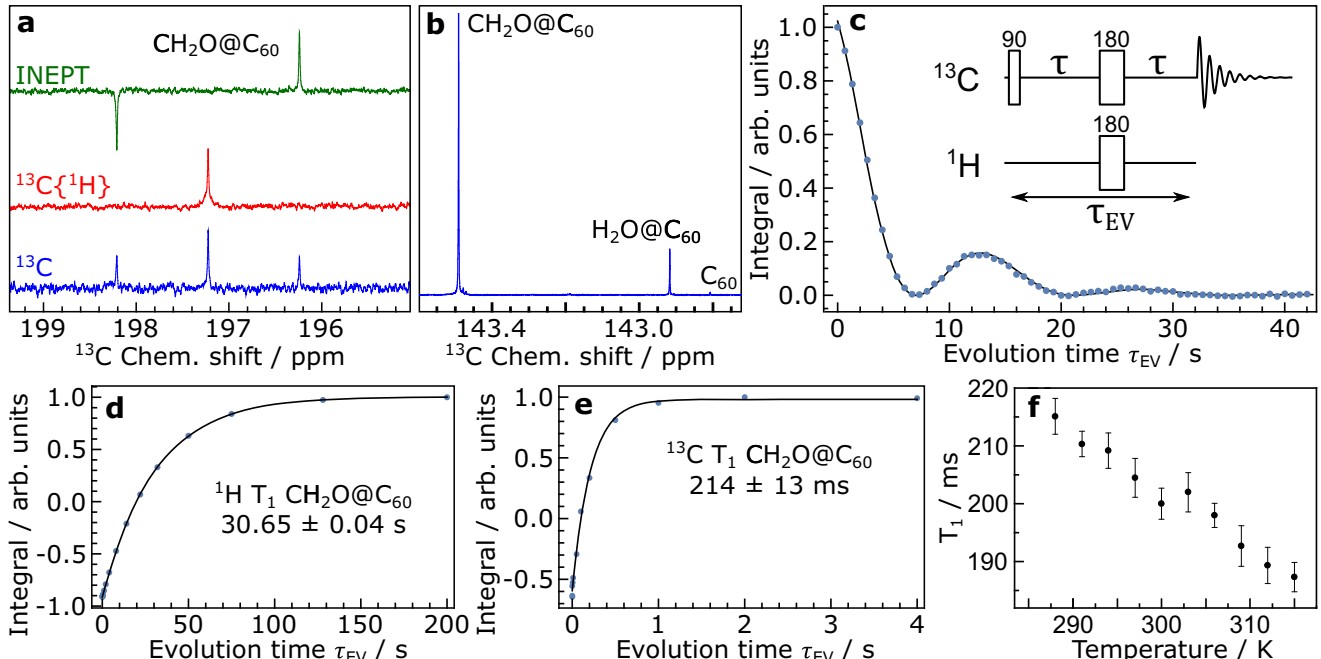

**Fig. 4 | $^{13}$C NMR of CH$_2$O@C$_{60}$.** Taken prior to complete removal of H$_2$O@C$_{60}$ and C$_{60}$ and sublimation, 23 mM in ODCB-$d_4$ at 16.45 T and 298 K (**a**–**e**). **a** showing the C̲H$_2$O triplet $^{13}$C spectrum, a proton decoupled $^{13}$C{$^1$H} spectrum and a non-refocused INEPT spectrum with inter-pulse delay of 1.44 ms. **b** expansion of the $^{13}$C spectrum around 143ppm, showing the $^{13}$C signals for CH$_2$O@C̲$_{60}$, H$_2$O@C̲$_{60}$ and empty C̲$_{60}$. **c** $^{13}$C (CH$_2$O@C̲$_{60}$) NMR signal amplitude modulation following the J-modulated spin-echo sequence shown in the figure [90($^{13}$C) – delay – 180($^{13}$C, $^1$H) –

delay – Acquire ($^{13}$C)], acquired with 16 transients. Fitting the modulation gives $|^0J_{HC}|$ = 70.6 ± 0.3 mHz. **d** and **e** Inversion-recovery curves for the T$_1$ spin-lattice relaxation time constant of CH$_2$O nuclei in CH$_2$O@C$_{60}$ for $^1$H and $^{13}$C (central line) respectively. **f** $^{13}$C T$_1$ of the central line of $^{13}$C-labelled CH$_2$O in $^{13}$CH$_2$O@C$_{60}$, measured at 16.45 T by inversion recovery as a function of sample temperature, in a 1 mM solution in toluene-$d_8$. The error bars represent Standard Error estimates in the fitted T$_1$ values.

Unoccupied Molecular Orbital (LUMO) gap is estimated to be 37% larger than this, i.e. −25 meV. Both Differential Pulse Voltametery (DPV) (Fig. 3b) and Cyclic Voltametery (CV) (Supplementary Fig. 18) show a decrease in the first 4 reduction potentials of CH$_2$O@C$_{60}$ by −30 mV relative to C$_{60}$ (Supplementary Table 4). The change in the 1$^{st}$ reduction potential indicates that the LUMO level is lowered in energy by 30 meV. Hence UV-vis and electrochemical studies indicate a reduction in the LUMO energy of 30 meV and increase in the HOMO energy of 5 meV of C$_{60}$ upon encapsulation of CH$_2$O. The incorporation of MeCN into the open fullerene **1** has recently been reported to lower its 1st oxidation potential by 40 meV[50]. Photoelectron spectroscopy of [H$_2$O@C$_{60}$]$^-$ indicates the electron affinity of H$_2$O@C$_{60}$ to be 8.8 meV higher than C$_{60}$[51] although no difference in UV spectra or cyclic voltammetry was observed[9].

### NMR studies on CH$_2$O@C$_{60}$
Detailed NMR experiments were performed at 16.45 T using a 23 mM solution of CH$_2$O@C$_{60}$ in 1,2-dichlorobenzene-$d_4$ (ODCB-$d_4$), degassed by bubbling with nitrogen gas for 10 min. The $^1$H chemical shift of the CH$_2$O@C$_{60}$ peak is 3.75 ppm. The $^{13}$C spectrum of CH$_2$O@C$_{60}$ shows a 1:2:1 triplet at 197.22 ppm with a J-coupling of $|^1J_{CH}|$ = 173.49 ± 0.09 Hz corresponding to the $^{13}$C peak of C̲H$_2$O@C$_{60}$ (Fig. 4a). The triplet collapses to a single peak when applying $^1$H decoupling. Polarisation transfer from $^1$H to $^{13}$C using an INEPT sequence with inter-pulse delay τ/2 = 1.44 ms ≃ |4J$_{HC}$|$^{-1}$ gives an antiphase $^{13}$C spectrum with enhanced outer peaks and a missing central peak, as expected[52].

A single peak is observed for the cage $^{13}$C of CH$_2$O@C̲$_{60}$ at 143.49 ppm, which is shifted by +0.684 ppm with respect to that of empty C$_{60}$ (Fig. 4b). The shift is substantially greater than that observed for CH$_4$@C$_{60}$ (+0.52 ppm)[11]. This conforms to the general finding that formation of an endofullerene leads to a change in the cage $^{13}$C chemical shift which increases with the size of the encapsulated species.

J-couplings between the nuclei of atoms which are not linked by a sequence of covalent bonds have been denoted $^0$J, where the superscript zero indicates the absence of a covalent linkage[26]. A $^0$J$_{HeC}$ coupling has been detected by a small splitting of the $^{13}$C resonance of the helium endofullerene $^3$He@C$_{60}$[26]. Neither the $^{13}$C cage peak of the CH$_2$O@C̲$_{60}$, nor the $^1$H peak of the majority $^{12}$C̲H$_2$O@C$_{60}$ isotopologue, display a resolved spectral splitting. Nevertheless, the presence of an unresolved $^0$J$_{HC}$-coupling is clearly indicated by an oscillation of the signal amplitude for the CH$_2$O@C̲$_{60}$ cage sites as a function of the evolution interval τ$_{EV}$, in a heteronuclear J-modulated spin-echo experiment (Fig. 4c). Since each $^{13}$C nucleus of the CH$_2$O@C̲$_{60}$ cage is coupled to the two $^1$H nuclei of C̲H$_2$O, the modulation of the NMR signal amplitude (the integral of the signal) oscillates at twice the J-coupling frequency. Fitting the modulation gives an estimated $^0$J-coupling $|^0J_{HC}|$ = 70.6 ± 0.3 mHz.

The solution-state $^1$H T$_1$ spin-lattice relaxation time constant of the endohedral CH$_2$O protons was measured by inversion recovery to be 30.65 ± 0.04 s, in ODCB-$d_4$ at 16.45 T and 298 K (Fig. 4d). It was only slightly shorter (28.0 ± 0.8 s at 298 K) in the $^{13}$CH$_2$O@C$_{60}$ isotopologue (Supplementary Fig. 21). This indicates that the relaxation rate contribution of the $^1$H–$^{13}$C dipole-dipole coupling mechanism is very small. The long $^1$H relaxation times are consistent with high rotational mobility for the endohedral molecule, which diminishes the contribution to relaxation from dipole-dipole and chemical shift anisotropy mechanisms.

A similarly long $^1$H T$_1$ relaxation time of 30.8 ± 0.8 s was observed for free monomeric CH$_2$O in a THF solution (Supplementary Fig 24). In contrast, the formaldehyde $^1$H relaxation in the open-cage system CH$_2$O@**2** was observed to be much faster: T$_1$ = 2.31 ± 0.03 s for CH$_2$O@**2** in ODCB-$d_4$ (Supplementary Fig. 23). The large difference between the $^1$H T$_1$ for the formaldehyde protons in CH$_2$O@C$_{60}$ and CH$_2$O@**2** is attributed to the low-symmetry confining environment in

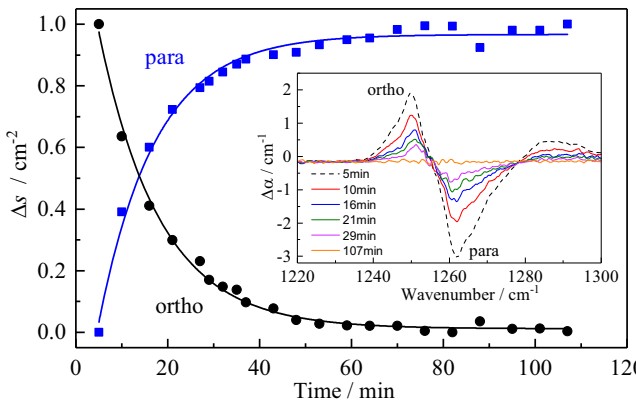

**Fig. 5 | Interconversion of ortho- and paraformaldehyde observed by infra-red spectroscopy.** Change of para and ortho species signal amplitude, $\Delta s$, measured at the 1255 cm$^{-1}$ ro-vibrational band of CH$_2$O@C$_{60}$ after the temperature jump from 20 K to 5 K. The data points are the normalised line areas integrated between 1235 and 1255 cm$^{-1}$ for the ortho spin isomers (black dots) and between 1255 and 1280 cm$^{-1}$ for the para spin isomers (blue squares) of CH$_2$O@C$_{60}$. The para signal grows with a time constant of $12.4 \pm 0.6$ min, while the ortho signal decays with a time constant of $13.3 \pm 0.4$ min at 5 K.

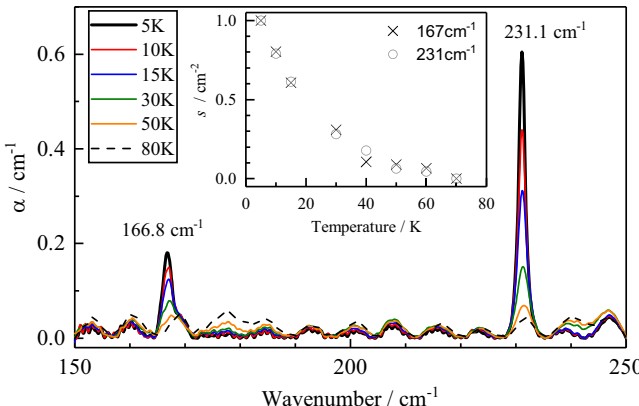

**Fig. 6 | Temperature dependence of the CH$_2$O@C$_{60}$ ($f$ = 1.0) far-infra-red (THz) absorption spectra.** Absorption coefficient as a function of wavenumber, between 5 K and 80 K. The translational modes are observed at 166.8 cm$^{-1}$ and 231.1 cm$^{-1}$. Inset: temperature dependence of the 167 cm$^{-1}$ (crosses) and 231 cm$^{-1}$ (open dots) integrated absorption peak area $s$, normalised to the peak area at 5 K, after subtracting the 80 K spectrum.

the latter case, which partially couples the formaldehyde orientation to that of the open-cage fullerene. Hence the long rotational correlation time of the open fullerene cage leads to a short $^1$H $T_1$ for the endohedral CH$_2$O in CH$_2$O@**2**. This effect is absent for the symmetrical CH$_2$O@C$_{60}$ complex, where the rotational motion of the formaldehyde guest is strongly decoupled from that of the fullerene host.

In contrast to the $^1$H relaxation, the $^{13}$C $T_1$ relaxation time constant for the endohedral formaldehyde in CH$_2$O@C$_{60}$ is surprisingly short: $214 \pm 13$ ms for natural abundance $^{13}$CH$_2$O@C$_{60}$ in ODCB-$d_4$ at 298 K (Fig. 4e) and $204 \pm 4$ ms for $^{13}$C-labelled $^{13}$CH$_2$O@C$_{60}$ in toluene-$d_8$ at 297 K (Fig. 4f). The $^{13}$C $T_1$ value for $^{13}$C-labelled $^{13}$CH$_2$O@C$_{60}$ decreases only slightly as the magnetic field is increased ($223 \pm 7$ ms at 9.4 T, $204 \pm 3$ ms at 14.1 T). This indicates that chemical shift anisotropy provides a relatively weak $^{13}$C relaxation mechanism in this case. However, the $^{13}$C $T_1$ has a strong inverse relationship on the sample temperature (Fig. 4f). This is indicative of a strong spin-rotation contribution to the $T_1^{-1}$ rate constant for the formaldehyde $^{13}$C nuclei. Strong spin-rotation relaxation is typical for nuclei of molecules in the gas phase[53]. This reinforces the general observation that molecules encapsulated in C$_{60}$ behave, in many respects, as if they are in the gas phase.

The strong spin-rotation relaxation of the $^{13}$C nuclei of formaldehyde, and the weakness of the same mechanism for the $^1$H nuclei, demands explanation. Experimental measurements of the spin-rotation tensors of isolated CH$_2$O molecules[54,55] indicate that the Frobenius norm of the $^{13}$C spin-rotation tensor is ~26 times larger than that of the $^1$H spin-rotation tensors. Since the relaxation rate constant induced by the spin-rotation mechanism, under plausible assumptions, is proportional to the square of the Frobenius norm of the relevant tensor[56], this explains the large ratio of the $^{13}$C to $^1$H $T_1^{-1}$ relaxation rate constants. A more detailed theory will require consideration of the anisotropic rotational motion of the formaldehyde inside the cage, the different orientations of the $^{13}$C and $^1$H spin-rotation tensors relative to the molecular reference frame, the inclusion of the dipole-dipole and chemical shift anisotropy relaxation mechanisms, and the possible influence of the fullerene cage on the spin-rotation interaction tensors of formaldehyde. This study is in progress and will be described elsewhere.

The $^{13}$C $T_1$ relaxation time constant for the cage $^{13}$C of CH$_2$O@C$_{60}$ was measured to be $17.42 \pm 0.06$ s in ODCB-$d_4$ at 298 K (Supplementary Fig. 22). This behaviour is unremarkable and is similar to that observed

for empty C$_{60}$[57]. The relaxation is attributed to a chemical shift anisotropy mechanism driven by rotational diffusion of the C$_{60}$ cages, which is not significantly affected by the endohedral molecule.

## IR and THz spectroscopy of CH$_2$O@C$_{60}$

**Ortho/para interconversion of CH$_2$O@C$_{60}$.** Spin-isomer conversion in CH$_2$O@C$_{60}$ was studied by a temperature jump method. The sample was allowed to equilibrate at 20 K and then rapidly cooled to 5 K. A series of IR spectra were taken at different times after the cooling. Subtraction of the spectrum measured two hours after the temperature jump from the spectra taken at earlier times gives a series of difference spectra in which the ortho-CH$_2$O@C$_{60}$ peaks are positive, while para-CH$_2$O@C$_{60}$ peaks are negative. The inset in Fig. 5 shows the difference spectra for a peak at 1235–1280 cm$^{-1}$, tentatively assigned as the $v_6$ (B$_2$) vibration. The points in Fig. 5 show the integrals of two different spectral regions as a function of time, one attributed to the para spin isomer (blue squares), and one attributed to the ortho spin isomer (black dots). The time constant for the ortho-para conversion is estimated to be approximately 13 min at 5 K.

**Quantised translational modes.** At temperatures below ~50 K, the far-infra-red (THz) spectrum of CH$_2$O@C$_{60}$ shows two sharp lines at 167 cm$^{-1}$ and 231 cm$^{-1}$ (Fig. 6). These lines do not appear in the far-IR spectra of C$_{60}$[58] nor He@C$_{60}$ (Supplementary Fig. 25). Both peaks get weaker with increasing temperature and have a similar temperature dependence, implying that they correspond to transitions starting from the ground state (inset to Fig. 6). Furthermore, a comparison of spectra taken at different times after cooling shows that the 167 cm$^{-1}$ peak has an unresolved structure, with components belonging to different spin isomers (Supplementary Fig. 26). These properties prove that the two peaks are indeed due to the endohedral CH$_2$O@C$_{60}$ molecules. However, the 167 and 231 cm$^{-1}$ lines do not correspond to vibrations of free CH$_2$O because all vibration frequencies of free CH$_2$O are above 1000 cm$^{-1}$[31]. Also, it is unlikely that the observed lines are rotational transitions of formaldehyde, because there are no rotational transitions of free CH$_2$O, starting from thermally populated low-lying states, in this spectral range[59]. We assign the 167 cm$^{-1}$ and 231 cm$^{-1}$ lines to translational transitions, i.e. transitions between the particle-in-a-box quantum levels of the confined CH$_2$O molecules. Translational peaks of this type have been observed previously for noble gas endofullerenes[17] and for H$_2$O@C$_{60}$[20]. We provisionally attribute the 231 cm$^{-1}$ peak to translation along the long axis of the molecule, where

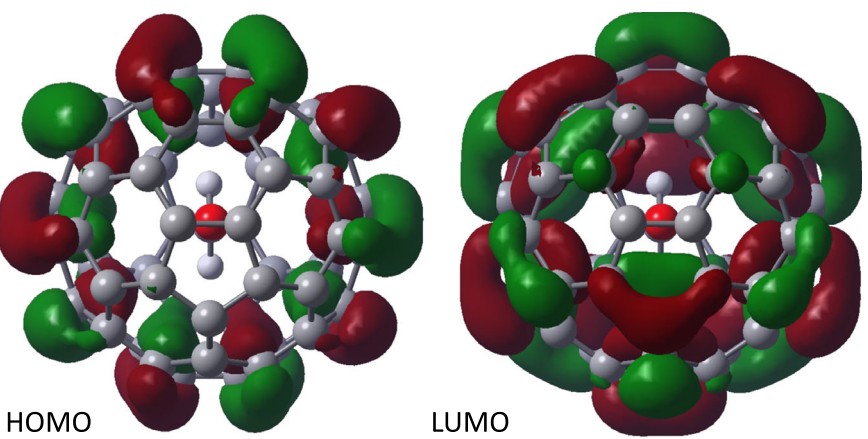

HOMO                                                                                          LUMO

**Fig. 7 | Molecular orbitals of CH$_2$O@C$_{60}$.** Shown at the minimum energy conformation calculated using DFT with B3LYP-D3 functional and cc-pVTZ basis set.

the confinement is particularly tight, and the 167 cm$^{-1}$ peak to translation perpendicular to the long axis of the molecule. A more detailed analysis of the THz and IR spectra of CH$_2$O@C$_{60}$ will be given elsewhere.

### Theoretical studies CH$_2$O@C$_{60}$

Some initial theoretical studies on CH$_2$O@C$_{60}$ were carried out using Density Functional Theory (DFT) to compare with the experimental results described above. Calculations were carried out with Gaussian 09 revision D1.01[60] using the B3LYP functional[61,62] with the Grimme D3 empirical dispersion correction using Beck–Johnson damping[63,64] and the cc-pVTZ basis set[65] with a superfine integration grid and tight criteria for convergence. We concentrate on differences in properties of the C$_{60}$ cage caused by incorporation of endohedral CH$_2$O, and on the translational modes resulting from its encapsulation.

In the minimised structure the C-O axis of the CH$_2$O aligns with the centre of the bond between 5 and 6 member rings of the cage (Fig. 7) although structures where it is aligned with the centre of the bond between two 6 member rings is only 0.1 kJ mol$^{-1}$ (1 meV) higher in energy (Supplementary Fig 19). The calculated parameters below for the second conformer are given in the Supplementary. Comparing the minimised structure with that of C$_{60}$, calculated using the same method, shows that the diameter of the cage measured along the C-O axis has increased by 3.8 pm (Supplementary Tables 7 and 8). The diameter in the plane of the CH$_2$O decreased by 0.2 pm, and perpendicular to the CH$_2$O plane decreased by 1.5 pm. The mean diameter increases by 0.6 pm.

The positions of the HOMO and LUMO of CH$_2$O@C$_{60}$ are strongly influenced by the presence of the endohedral molecule (Fig. 7). The LUMO energy of CH$_2$O@C$_{60}$ is calculated to be 35 meV lower than C$_{60}$ which compares well to the 30 mV reduction in the first reduction potential found electrochemically (Supplementary Table 9). The HOMO is little changed (+3.5 meV) giving a calculated HOMO-LUMO gap for CH$_2$O@C$_{60}$ 38 meV smaller than C$_{60}$ (cf. 25 meV reduction estimated experimentally from the difference in optical band gaps above). The excitation energy to the 1st excited states of CH$_2$O@C$_{60}$ and C$_{60}$ were calculated using TD-DFT (B3LYP-D3/cc-pVTZ) and the former found to be 23.8 meV smaller (Supplementary Table 11) corresponding to a 6.9 nm increase wavelength of the longest wavelength absorption in the UV-vis spectra, in excellent agreement with that observed (6 nm change).

Frequency calculations on the minimised structure of CH$_2$O@C$_{60}$ gave three translational modes. That corresponding to movement along the long axis of CH$_2$O is at 233 cm$^{-1}$, surprisingly good agreement with the observed peak at 231 cm$^{-1}$ (Supplementary Table 10). Calculations also give translational modes in the plane of the CH$_2$O at 202 cm$^{-1}$, and perpendicular to that plane at 175 cm$^{-1}$, compared to the

observed single peak at 167 cm$^{-1}$. DFT neglects the delocalised wave nature of nuclei, treating them as fixed points (Born-Oppenheimer approximation) whereas the observation of ortho and para hydrogen spin isomers due to rotation about the C-O axis indicate that the hydrogen nuclei are delocalised, even at cryogenic temperatures, so the observation of a single peak for translation perpendicular to the C-O axis is reasonable.

The $^{13}$C chemical shift of the cage carbons in CH$_2$O@C$_{60}$ was calculated using the Gauge-Independent Atomic Orbital (GAIO) method to be 0.80 ppm greater than that of empty C$_{60}$ (Supplementary Tables 12 and 13). This is in rough agreement with the observed 0.684 ppm increase in chemical shift of the cage $^{13}$C upon encapsulation of CH$_2$O. Since the DFT calculations are performed at 0 K in the gas phase, a difference to room temperature liquid state NMR data is to be expected. Encapsulation of CH$_2$O by C$_{60}$ may change the $^{13}$C chemical shift of the cage carbons by at least two separate mechanisms: (i) direct interactions with the electrons and nuclei of the guest molecule, and (ii) expansion of the cage geometry upon accommodation of the guest, which in turn modifies the electronic structure of the cage and hence the $^{13}$C chemical shift. The relative importance of these contributions was assessed as follows: First, calculations were performed on empty C$_{60}$, but fixing the geometry of the C$_{60}$ cage to that of CH$_2$O@C$_{60}$ (as described above). In this case, the calculated $^{13}$C chemical shift of the cage carbons was found to be 0.35 ppm greater than that of empty C$_{60}$ with the energy-minimised geometry. Second, calculations were performed on CH$_2$O@C$_{60}$, but fixing the geometry of the C$_{60}$ cage to that of empty C$_{60}$. In this case, the calculated $^{13}$C chemical shift of the cage carbons was 0.45 ppm greater than that of empty C$_{60}$ with the energy-minimised geometry. We conclude that the direct interactions with the guest molecule, and the geometry changes in the C$_{60}$ cage, both contribute to the observed 0.684 ppm change in the $^{13}$C chemical shift of the cage carbons in CH$_2$O@C$_{60}$, relative to that of C$_{60}$.

In summary, formaldehyde CH$_2$O was successfully encapsulated into C$_{60}$ despite the large nominal size of formaldehyde relative to the C$_{60}$ cavity. The large size of the CH$_2$O significantly inhibited several orifice contraction steps. The isotopically labelled materials $^{13}$CH$_2$O@C$_{60}$ and CD$_2$O@C$_{60}$ were also made.

A significant perturbation of the electronic structure of C$_{60}$ is observed in CH$_2$O@C$_{60}$. UV-vis spectroscopy shows a 6 nm red shift in the longest wavelength absorption, and electrochemistry a 30 meV lowering of the 1st reduction potential on incorporation of CH$_2$O into C$_{60}$. However, X-ray crystallography did not show a significant change in C$_{60}$ geometry.

The solution-state NMR spectroscopy of CH$_2$O@C$_{60}$ displays a confinement-induced 70 mHz J-coupling between the cage $^{13}$C nuclei and the formaldehyde $^1$H nuclei. The spin-lattice relaxation behaviour

is very unusual, with the $^{13}C$ nuclei of formaldehyde relaxing about 150 times faster than the $^{1}H$ nuclei. The rapid $^{13}C$ relaxation is attributed to a strong spin-rotation relaxation mechanism, associated with unusually free rotation.

The tightly confined $CH_2O$ molecules display spatial quantisation, with two translational modes observed in the far-infra-red, at 167 cm$^{-1}$ and 231 cm$^{-1}$. Despite this tight confinement, the encapsulated formaldehyde molecules rotate freely about their long axes. Conversion between the ortho and para nuclear spin isomers is observed at low temperatures by infra-red spectroscopy, with a spin-isomer conversion time constant of ~13 min at 5 K. To the best of our knowledge, this is the first time that the spin-isomer conversion of formaldehyde has been observed in a condensed phase.

DFT calculations on $C_{60}$ and $CH_2O@C_{60}$ predict that incorporating the endohedral species distorts the cage by +3.8 pm along the long axis of the $CH_2O$, although the average diameter, which might be expected to be observed for a freely rotating endohedral molecule, only increases by 0.6 pm. Calculations are consistent with the experimentally observed changes in the electrochemical reduction potentials, the UV absorption spectra, and the $^{13}C$ chemical shift of the cage carbons. The frequencies of the translational modes of the confined $CH_2O$ molecules predicted by DFT calculations are in reasonable agreement with the experimental results.

## Methods

### Synthesis

**General methods**. Toluene was freshly distilled from sodium benzophenone ketal under argon. MeCN was freshly distilled from CaH$_2$ under argon. Technical grade 1-chloronaphthalene (≥85%) was filtered through a short column of activated Al$_2$O$_3$ and distilled under argon at reduced pressure (117 °C, 13 mbar). Triisopropyl phosphite was distilled over sodium at reduced pressure (67 °C, 14 mbar). Dimethyldioxirane was prepared according to the published procedure[43]. C$_{60}$ (98%) was purchased from Solaris Chem Inc. All other reagents or solvents were used as received from commercial suppliers. Open-cage fullerene **1** was prepared according to the published methods[7,9].

### CH$_2$O@2

Finely ground sulfide **1** (438 mg, 386 mmol) was dissolved in dry, oxygen-free THF (40 mL) in a round bottom flask with an inlet tube connected to a round bottom flask containing excess paraformaldehyde (8.15 g) and $p$-toluene sulfonic anhydride (2.09 g), and heated to 50 °C. The flask containing paraformaldehyde was heated using a hot-air gun and the monomeric formaldehyde produced was bubbled through the sulfide **1** solution for 30 min using a flow of N$_2$ gas[43]. After cooling to room temperature, THF was removed by rotary evaporation and the crude $CH_2O$ containing sulfide **1** was dissolved in toluene and quickly purified using column chromatography (SiO$_2$ eluted with a 90:8:2 mixture of toluene:EtOAc: AcOH).

To a stirred solution of the mixture of $CH_2O@$**1** and $H_2O@$**1** + **1** in toluene (80 mL) at 0 °C, was added dimethyldioxirane (10.0 mL of a 98.0 mM solution in acetone, 1.00 mmol) rapidly using an ice-chilled syringe. The resulting mixture was stirred at 0 °C for 10 min. before removal of the cooling bath and stirring for 1 h. Solvents were removed in vacuo to give the title compound as a crude brown powder containing $CH_2O@$**2** and $H_2O@$**2** + **2** which was used directly in the next step without purification (403 mg, 0.341 mmol, 88%). The fillings in **2** was determined by NMR as 70% $CH_2O@$**2** and 30% $H_2O@$**2** + **2**.

### $^{13}$CH$_2$O@2 and CD$_2$O@2

Sulfide **1** (478 mg, 0.421 mmol) was finely ground with $^{13}C$ paraformaldehyde (99 mg) and loosely packed in a 1/4″ stainless steel tube. The tube was sealed at both the ends using Swagelok caps and kept in preheated oven at 105 °C for three hours. After cooling to room temperature the tube was opened and the crude material quickly dissolved in toluene (40 ml). The crude solution was passed through a silica column using 90:8:2 (Toluene: EtOAc: AcOH) as an eluent. The purified material was concentrated on rotary evaporator and dissolved in toluene (40 mL). To this solution was added dimethyldioxirane (10.0 mL of a 98.2 mM solution in acetone, 1.00 mmol) rapidly using an ice-chilled syringe. The resulting mixture was stirred at 0 °C for 10 min. before removal of the cooling bath and stirring for 1 h. Solvents were removed in vacuo to give the mixture of $^{13}$CH$_2$O@**2** and $H_2O@$**2** + **2** as a brown powder which was used directly in the next step without purification (470 mg, 0.398 mmol, 95%). The filling of $^{13}$CH$_2$O@**2** was determined as 24% by integrating the alkene peaks of $^{13}$CH$_2$O@**2** and $H_2O@$**2** + **2** in the $^{1}H$ NMR spectrum.

The same method was used to synthesise $CD_2O@$**2** using tetraketone **1** (430 mg, 0.379 mmol), deuterated paraformaldehyde (98 mg), dimethyldioxirane (10.0 mL of a 98.2 mM solution in acetone, 1.00 mmol) and Toluene (40 mL). The mixture of $CD_2O@$**2** and $H_2O@$**2** + **2** was obtained as a brown powder which was used directly in the next step without purification (383 mg, 0.324 mmol, 86%). The filling of $CD_2O@$**2** was determined as 25% by integrating alkene peaks in the $^{1}H$ NMR of $CD_2O@$**2** and $H_2O@$**2** + **2**.

### CH$_2$O@4

A purpose-built reaction vessel (Supplementary Fig. 29) was charged with mixture of $CH_2O@$**2** and $H_2O@$**2** (380 mg, 0.322 mmol) and the apparatus placed under an atmosphere of argon. THF (250 mL, degassed), AcOH (50 mL of a degassed 10% v/v aqueous solution) and toluene (250 mL) were added, and the resulting mixture was vigorously stirred under irradiation with a 3 × 100 W yellow (590–595 nm) LED lamp, for 18 h at 55 °C. Solvents were then removed in vacuo. Purification by rapid, repeat column chromatography (SiO$_2$ eluted with a 90:8:2 mixture of toluene:EtOAc:AcOH; then SiO$_2$ eluted with 2% AcOH in toluene) gave mixture of $CH_2O@$**3** and $H_2O@$**3** as a dark brown solid (130 mg, 0.115 mmol, 36%) which was used directly in the next step.

The mixture of $CH_2O@$**3** and $H_2O@$**3** obtained above (130 mg, 0.115 mmol) was transferred to the RBF and was back filled with N$_2$ and triphenylphosphine (310 mg, 1.18 mmol) and dry toluene was added under N$_2$ and the resulting mixture stirred for 72 h at reflux. After cooling to room temperature, solvents were removed in vacuo. Purification by column chromatography (SiO$_2$ eluted with a gradient of 1:1 hexane:toluene → toluene) gave mixture of $CH_2O@$**4** and $H_2O@$**4**, as a brown/black solid (104 mg, 0.0945 mmol, 82%). 35% filling of $CH_2O@$**4** was determined by comparison of integrals in the experimental $^{1}H$ NMR spectrum.

The labelled materials $^{13}$CH$_2$O@**4** and $CD_2O@$**4** were prepared in the same way.

### Synthesis of CH$_2$O@C$_{60}$

Into a dry flask containing 25% filled $CH_2O@$**4** (rest $H_2O@$**4** + **4**) (210 mg, 0.191 mmol) was added toluene (25 mL) and dry, distilled P(OPr$^i$)$_3$ (760 µL, 3.22 mmol). The solution was heated to reflux in the dark for 16 h. The reaction mixture was poured directly onto a SiO$_2$ column and eluted with toluene to collect a purple band. Removing the solvent *in vacuo* afforded a mixture of $CH_2O@$**5** and $H_2O@$**5** (22% filling of $CH_2O$ measured by NMR) which was immediately dissolved in 1-chloronaphthalene (43 mL) and transferred to a Young's tube containing N-phenyl maleimide (88 mg, 0.51 mmol). The solution was degassed under dynamic vacuum for 10 min, put under N$_2$ atmosphere and sealed. The flask was immersed into a preheated metal heating block at 255 °C and stirred for 40 h. After cooling to room temperature, the solution was flushed through a SiO$_2$ column packed with toluene, collecting the required product

as a purple band, followed by a red band containing the side product **6**. The side product $CH_2O@\textbf{6} + H_2O@\textbf{6}$ was further purified using hexane: EtOAc 90:10 to give 39 mg of 37% filled $CH_2O@\textbf{6}$, see data for the same in 1.4. After removing the bulk of the toluene *in vacuo* the remaining toluene and 1-chloronaphthalene were distilled off under vacuum (<1 torr). The crude product was purified by preparative HPLC (2 × 20 mm × 250 mm) Cosmosil™ Buckyprep columns in series eluting with toluene at a flow rate of 10 mL min$^{-1}$ to remove traces of 1-chloronapthalene to give a mixture of $C_{60}$ (4.5%), $H_2O@C_{60}$ (80%) and $CH_2O@C_{60}$ (15.5%), 66 mg, 46%, 0.088 mmol (ratios from HPLC trace). The mixture again subjected to recycling HPLC using the same conditions (retention time after 7 cycles: $C_{60}$: 220.3 min; $H_2O@C_{60}$: 228.9 min $CH_2O@C_{60}$: 238.3 min) to give **$CH_2O@C_{60}$** (>99% filled, remainder $H_2O@C_{60}$, 7.1 mg, 0.0093 mmol, 20% based on the estimated amount of $CH_2O@\textbf{4}$ in the starting material).

The labelled materials $^{13}CH_2O@C_{60}$ and $CD_2O@C_{60}$ were prepared in the same way.

### $CO_2$@Sulfoxide
Finely grounded **1** (402 mg, 0.354 mmol) was put into a stainless steel tube (100 mm, 13.2 mm o.d., 5.2 mm i.d) with SITEC® high-pressure fittings, and loosely plugged with glass wool. The tube was heated at 180 °C for 2 h at <1 torr. After cooling to room temperature the tube was filled with $CO_2$ to 50 bar using a SITEC® 750.01 hand-operated pressure intensifying syringe. The reactor was then heated to 100 °C and maintained at this temperature for 18 h, with a stable internal pressure of 58–59 atm. The reactor was cooled after 18 h and pressure was released slowly, the tube was taken out and material was collected as dark red solid which was directly taken for DMDO oxidation. To a stirred solution of $CO_2$@**1** in toluene (40 mL) at 0 °C, was added dimethyldioxirane (10 mL of a 98.2 mM solution in acetone, 1.0 mmol) rapidly using an ice-chilled syringe. The resulting mixture was stirred at 0 °C for 10 min. before removal of the cooling bath and stirring for 1 h, during which time the mixture warmed to room temperature. Solvents were removed in vacuo to give the title compound as a crude brown powder (381 mg, 0.319 mmol, 90%) which was used directly in the attempted photochemical orifice contraction without purification. The filling factor was determined by comparison of integrals in the experimental $^1H$ NMR spectrum.

### Density functional theory calculations
Binding constants and activation energies for entry/loss of endohedral species into sulfide **1** and sulfoxide **2** were determined using model structures **1a** and **2a** in which the 6-tert-butylpyridyl groups were replaced by methyl substituents. Calculations were carried out using Gaussian 09[60], using either the B3LYP functional[61,62] with Grimme D3 empirical dispersion correction using Beck–Johnson damping[63,64], or the M06-2X functional[66] with cc-pVDZ basis set[65] to locate minimum energy and transition state structures and to characterise them through frequency calculations. The cc-pVTZ[65] basis set with an ultrafine integration grid was used to calculate electronic energies and to correct for Basis Set Superposition Error using the counterpoise method[67]. Thermal corrections to the electronic energy to give the free energy (G) at 298 K and 1 atm were derived from frequency calculations at the cc-pVDZ level using the Gaussian freqchk utility[68]. The frequencies were not scaled and low frequency modes were not removed. The values from B3LYP are used in the paper, those from M06-2X (also widely used in the published literature for calculations on endohedral open fullerenes) are given for comparison in Supplementary Tables 5 and 6.

Calculations on $CH_2O@C_{60}$ were carried out using the B3LYP functional[61,62] with Grimme D3 empirical dispersion correction using Beck–Johnson damping[63,64], and the cc-pVTZ basis set[65] with a superfine integration grid. Minimum energy structures were located using

the tight convergence criteria and characterised with frequency calculations. Details on geometry, orbital energy levels, excited state energies, predicted $^{13}C$ NMR shifts and calculated vibrations and rotations are given in Supplementary Tables 7−13.

### X-ray crystallography
Dark orange plate-shaped crystals of the nickel(II) octaethylporphyrin/benzene solvate of $CH_2O@C_{60}$ were obtained from benzene by slow evaporation. A suitable crystal with dimensions $0.08 × 0.08 × 0.05$ mm$^3$ was selected and mounted on a MITIGEN holder with silicon oil on a ROD, Synergy Custom system, HyPix diffractometer. The crystal was kept at a steady T = 103(1) K during data collection. The structure solved and the space group P−1 (# 2) determined with the ShelXT 2014/5 solution programme[69] using dual methods and by using Olex2 1.5-alpha[70] as the graphical interface. The model was refined with ShelXL 2016/6[70,71] using full matrix least squares minimisation on F2 constraining the $CH_2O$ molecule to idealised geometry.

### Voltammetry of $CH_2O@C_{60}$
Solutions of ~ 3 mg of $C_{60}$ or ~1.5 mg $CH_2O@C_{60}$ were prepared in 5 mL of a 4:1 mixture of toluene and acetonitrile containing 0.1 M $Bu_4N.BF_4$ as electrolyte. The cell contained a 3 mm diameter glassy carbon working electrode, a 1 cm$^2$ sheet of platinum as the counter electrode and a silver wire pseudo-reference electrode. Cyclic Voltammetry (CV) and Differential Pulse Voltammetry (DPV) were carried out using an AUTOLAB PG204 potentiostat at room temperature using a scan rate of 50 mV/s. After acquiring initial scans of each component, 0.1 mL of a 3 mg/mL solution of ferrocene in toluene was added as an internal reference and further scans acquired.

### NMR studies on $CH_2O@C_{60}$
Detailed NMR experiments were performed at 16.45 T, carried out using a Bruker Ascend 700 NB magnet fitted with a Bruker AVANCE NEO console and Bruker TCI prodigy 5 mm liquids cryoprobe. An approx. 23 mM solution of $CH_2O@C_{60}$ in 1,2-dichlorobenzene-$d_4$ (ODCB-$d_4$) was prepared by dissolving 14 mg of powdered $CH_2O@C_{60}$ (86.7% filling) in 0.8 mL of the solvent. The solution was then filtered and then degassed by bubbling nitrogen gas through the solution for 10 min. $^1H$ and $^{13}C$ NMR spectra were referenced to the solvent chemical shifts (1,2-dichlorobenzene-$d_4$), $^1H$ = 6.93 ppm and $^{13}C$ = 127.19 ppm[7]. All chemical shifts have confidence limits of ±0.01 ppm. NMR measurements on $^{13}C$-labelled $^{13}CH_2O@C_{60}$ were performed at 16.45 T on a 1 mM solution in toluene-$d_8$, degassed by bubbling with nitrogen gas for 10 min.

### IR and THz spectroscopy of $CH_2O@C_{60}$
A sample of $CH_2O@C_{60}$ which had been purified to 100% filling by recirculating HPLC was sublimed (550 °C, 10$^{-5}$ bar) before the measurements. The sublimed powder was pressed into pellet 3 mm in diameter and with thickness d = 0.355 mm. IR spectroscopy measurements were carried out on a Bruker Vertex 80 v spectrometer. Between 30 and 600 cm$^{-1}$ a 4 K bolometer detector and above 600 cm$^{-1}$ a mercury cadmium telluride detector with glowbar light source were used. Instrument resolution was 0.3 cm$^{-1}$. The sample pellet was mounted inside a vacuum-tight sample chamber filled with helium exchange gas and attached to a cold finger of a continuous flow cryostat. The absorption coefficient was calculated from the intensities transmitted through the sample and through a 3 mm diameter reference hole.

### Data availability
Supplementary material containing spectroscopic and analytical data and copies of $^1H$ and $^{13}C$ NMR spectra of synthesised compounds; Kinetic data on loss of $CH_2O$ from **1** and **2**, and $CO_2$ from **2**; Additional

plots and data from UV, voltametry, NMR and IR studies, and energies and measurements of species from DFT calculations are provided in the supplementary information file. Cartesian coordinates from DFT calculations are provided in the source data file. Crystallographic data for the structure reported in this article has been deposited at the Cambridge Crystallographic Data Centre under deposition no. CCDC 2126579. This data can be obtained free of charge via www.ccdc.cam.ac.uk/data_request/cif, or by emailing data_request@ccdc.cam.ac.uk, or by contacting The Cambridge Crystallographic Data Centre, 12 Union Road, Cambridge ■CB2 1EZ, UK; fax: +44 1223 336033. All other data are available from the corresponding authors upon request. Source data are provided with this paper.

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

## Acknowledgements

This research was supported by the EPSRC (UK) under Grant Nos. EP/P009980/1 and EP/T004320/1 (MHL and RJW), the Estonian Ministry of Education, personal research funding PRG736 (UN), and the European Regional Development Fund, Project No. TK134 (UN). RJW acknowledges the use of the IRIDIS High Performance Computing Facility at the University of Southampton. We thank James W. Whipham for discussions.

## Author contributions

VKV and ESM performed synthetic experiments; GRB and MS performed NMR experiments; TJ, AS and UN performed IR / THz experiments. RJW performed DFT theoretical studies; MEL solved the crystal structure. RJW, MHL and TR designed and supervised the project. All authors contributed to writing the manuscript and have given approval to the final version.

## Competing interests

The authors declare no competing interests.
