## [Peer Review File · Nature Communications]

Squeezing formaldehyde into C60 fullereneReviewers' Comments:

Reviewer #1:

Remarks to the Author:

In this manuscript, the authors report on the preparation of the H₂CO@C₆₀ endohedral complex, the crystal structure of its nickel(II) octaethylporphyrin/benzene solvate, its UV/VIS, ¹H and ¹³C NMR, IR (including the far-IR region) spectra, and its electrochemical properties measured with DPV and CV.

Placing the bulky H₂CO> molecule inside the C₆₀ cage is a remarkable experimental achievement of significant importance to various branches of chemistry and physics as only a few endohedral complexes of the C₆₀ fullerene with guests comprising more than two atoms have been characterized thus far and their number is expected to remain limited. In light of this fact, Nature Communications appears to be a proper medium for the dissemination of the results of this cutting-edge research. The synthetic methodology reported by the authors in detail is sound and should be readily reproducible. The same goes for the reported measurements of the properties of the new endohedral complex.

As for the revisions aimed at improving the manuscript, there are two options. Should the authors aspire to make their paper more complete, the inclusion of the experimentally determined dipole moment of H₂CO@C₆₀ (and its comparison with that of the host in vacuo) would achieve this goal. In fact, this is the major revision I recommend provided the authors are able to complete the necessary experiment in short time (which should not be difficult). In addition, there are a few minor points that should be addressed:

1. The statement "This conforms to the general finding that formation of an endofullerene leads to a change in the cage ¹³C chemical shift which increases with the size of the encapsulated species." is an oversimplification. These changes stem from the influence of the guests on the electronic structure of the host cage. Whereas they are expected to increase with decreasing distances between the atoms of the guest and those of the host, the magnitude of this effect depends not only on the spatial extent of the guest but also on the detailed characteristics of its electron density.
2. The statement "We provisionally attribute the 231 cm⁻¹ peak to translation along the long axis of the molecule, where the confinement is particularly tight, and the 167 cm⁻¹ peak to translation perpendicular to the long axis of the molecule." requires further elaboration. In this context, quoting the results of simple DFT calculations involving vdW-including functionals would be particularly helpful. Such calculations can be carried out without much computational effort.
3. The word "supermolecular" is used only once in the text, the more accurate term "endohedral" appearing elsewhere. For the sake of uniformity, it should be replaced accordingly.

In summary, the paper is publishable upon incorporation of three minor revisions and, preferably, upon the addition of the dipole moment data.

Reviewer #2:

Remarks to the Author:

This manuscript describes the synthesis and properties of novel molecule-encapsulating fullerene C₆₀, i.e., HCHO@C₆₀, which is a novel compound that would attract much attention from many research fields. A molecule of HCHO was introduced inside the cage-opened C₆₀ derivative, whose sulfide moiety was oxidized to sulfoxide (compound 2). Photochemical reaction on compound 2 followed by treatments with PPh₃, P(OPr)₃, and N-phenylmaleimide gave HCHO@C₆₀. The structure was determined by the single crystal X-ray analysis, and absorption and redox properties were revealed. Furthermore, NMR chemical shifts, relaxation properties, and far-infrared spectra were discussed.

Although the synthetic method seems extension of their previous works, the final compound is very interesting. Thus, the reviewer recommends publication after revision described below.

- The calculated energies for entry and escape of the small molecules were shown in the main text and Extended Data Table 1. However, the similar data has been already reported in ref. 42 & 44. It is necessary to show that the present data are similar to the reported data or different from them, with citation of the previous data and comments based on comparison with them. In the Extended Data Table 1, please replace "<-" with other description to show chemical equilibrium.
- In the synthetic part in the main text and Figure 1, please make it clear whether the reaction conditions and yields of the products are similar or different from those for the previous examples. It would be helpful if such comparison is clearly provided including the conditions and yields for the synthesis of H₂O@C₆₀, HF@C₆₀, CH₄@C₆₀, Ar@C₆₀, and Kr@C₆₀, to see if the size of HCHO gives some influence or not.
- The filling factors and the chemical yields of the final step of closing for HCHO-encapsulating materials, CO₂-encapsulating material, and labelled materials are rather difficult to understand. It is highly recommended to show the results using chemical scheme with exact numbers of the chemical yields and filling factors. It would be very nice if some comments and data are added to explain the results based on the stabilization/destabilization energies by DFT calculations.
- The X-ray analysis is not fully completed. It is needed to apply disorder models toward the HCHO moiety, which would sit in different orientations with appropriate disorder factors.
- Regarding the redox potentials, the authors need careful check on all the results: Although the first reduction of HCHO@C₆₀ is shown to be more negative in Figure 3, however, it is less negative in Extended Data Fig.3, Extended data Table 2, and main text. If the more negative redox potential for HCHO@C₆₀ is correct, the story of main text should be largely modified. It would be possible to add discussion on the LUMO level obtained by DFT calculations.
- Although the T₁ value of the ¹H signal for HCHO@C₆₀ is reported to be extremely long such as 30 s, discussion on it looks too short. It would be interesting to compare the data with those for free HCHO dissolved in ODCB, HCHO@2, HCHO@3, HCHO@4, and HCHO@5 under the same temperature and magnetic field, to see if confinement effects are major factor or not.

Reviewer #3:

Remarks to the Author:

This is an interesting manuscript which describes the synthesis of a new endohedral fullerene, namely CH₂O@C₆₀ where a polyatomic formaldehyde molecule is located inside the C₆₀, by using the molecular surgery strategy. Interestingly, the formaldehyde molecule fits within the C₆₀ internal cavity (3.7 Å), despite the larger length of the CH₂O molecule (4.38 Å). The presence of the molecule significantly affects some features of the hosting molecule of fullerene such as the HOMO-LUMO gap and spin-spin coupling between the ¹³C nuclei of C₆₀ and the ¹H nuclei of formaldehyde, as well as in the own formaldehyde molecule, exhibiting a relaxing time for the ¹³C nuclei of formaldehyde 150 times faster than the ¹H nuclei. Furthermore, a nature of ¹particle in a has been confirmed according to the observation of two quantised translational modes in the THz spectra under cryogenic conditions. In summary, the authors have carried out a complex and skilful synthesis of the new endohedral and the further experiments have led to interesting conclusions. The results obtained in this study are sound and allow the advance of this interesting family of endohedrals of C₆₀ by molecular surgery. Therefore, I feel that this work meets the criteria of novelty and quality to be accepted for publication in Nat. Commun. after addressing the following minor points:

1. The authors should comment if the presence of the nickel(II) octaethylporphyrin in the X-ray study has some effect on the orientation of the inner CH₂O molecule.
2. Although provided in the methods section, the electrochemical experimental conditions, namely electrode and solvent used should be included in the main text (or alternatively in the figure caption). In this regard, I wonder if the presence of the formaldehyde carbonyl group is the responsible for this

LUMO decrease observation, or simply the proximity of the oxygen atom.

3. An in dept NMR study has been carried out with amazing results. In this regard, the study confirms that formaldehyde molecule behaves as in the gas phase. However, the authors could provide an insight to the observation that the ^{13}C nuclei of the encapsulated formaldehyde have a strong spin-rotation relaxation mechanism, while the ^1H nuclei do not.

4. The spin-isomer conversion of formaldehyde has never been observed experimentally in the condensed phase. However, in this condensed phase there is not interaction in between formaldehyde molecules. Although I am not an expert in this field, I wonder if it is fair to say condensed phase.

Response to REVIEWER COMMENTS

Reviewer #1 (Remarks to the Author):

As for the revisions aimed at improving the manuscript, there are two options. Should the authors aspire to make their paper more complete, the inclusion of the experimentally determined dipole moment of $\text{H}_2\text{CO}@\text{C}_{60}$ (and its comparison with that of the host in vacuo) would achieve this goal. In fact, this is the major revision I recommend provided the authors are able to complete the necessary experiment in short time (which should not be difficult).

Measuring the dipole moment is a very substantial piece of work, and unfortunately not something we can do with the amount of material, or the personnel resource available.

In addition, there are a few minor points that should be addressed:

1. The statement "This conforms to the general finding that formation of an endofullerene leads to a change in the cage ^{13}C chemical shift which increases with the size of the encapsulated species." is an oversimplification. These changes stem from the influence of the guests on the electronic structure of the host cage. Whereas they are expected to increase with decreasing distances between the atoms of the guest and those of the host, the magnitude of this effect depends not only on the spatial extent of the guest but also on the detailed characteristics of its electron density.

It is of course an oversimplification which is why the statement was worded as it is – we comment on the correlation to size, not the reason for it. Since it is of interest we have added the results of calculations of NMR shifts which demonstrate (within the Born-Oppenheimer approximation limitation of DFT) that the effect is roughly equal divided between the effect of the endohedral molecule on the size of the cage, and the effect of the endohedral molecule electrons.

"The ^{13}C chemical shift of the cage carbons in $\text{CH}_2\text{O}@\text{C}_{60}$ was calculated using the Gauge-Independent Atomic Orbital (GAIO) method to be 0.80 ppm greater than that of empty C_{60} (Supplementary Tables 12 and 13). This is in rough agreement with the observed 0.684 ppm increase in chemical shift of the cage ^{13}C upon encapsulation of CH_2O . Since the DFT calculations are performed at 0 K in the gas phase, a difference to room temperature liquid state NMR data is to be expected. Encapsulation of CH_2O by C_{60} may change the ^{13}C chemical shift of the cage carbons by at least two separate mechanisms: (i) direct interactions with the electrons and nuclei of the guest molecule, and (ii) expansion of the cage geometry upon accommodation of the guest, which in turn modifies the electronic structure of the cage and hence the ^{13}C chemical shift. The relative importance of these contributions was assessed as follows: First, calculations were performed on empty C_{60} , but fixing the geometry of the C_{60} cage to that of $\text{CH}_2\text{O}@\text{C}_{60}$ (as described above). In this case, the calculated ^{13}C chemical shift of the cage carbons was found to be 0.35 ppm greater than that of empty C_{60} with the energy-minimized geometry. Second, calculations were performed on $\text{CH}_2\text{O}@\text{C}_{60}$, but fixing the geometry of the C_{60} cage to that of empty C_{60} . In this case, the calculated ^{13}C chemical shift of the cage carbons was 0.45 ppm greater than that of empty C_{60} with the energy-minimized geometry. We conclude that the direct interactions with the guest molecule, and the geometry changes in the C_{60} cage, both contribute to the observed 0.684 ppm change in the ^{13}C chemical shift of the cage carbons in $\text{CH}_2\text{O}@\text{C}_{60}$, relative to that of C_{60} ."

2. The statement "We provisionally attribute the 231 cm^{-1} peak to translation along the long axis of the molecule, where the confinement is particularly tight, and the 167 cm^{-1} peak to translation perpendicular to the long axis of the molecule." requires further elaboration. In this context, quoting the results of simple DFT calculations involving vdW-including functionals would be particularly helpful. Such calculations can be carried out without much computational effort.

We have carried out such calculations and have added them to the paper (with surprisingly excellent agreement for the 231 cm^{-1} peak). As expected (Born-Oppenheimer approximation again) calculations give two quantised translations perpendicular to the C-O axis but the average is in moderate agreement with the observed peak. A proper analysis of rotation-vibration mixing is of course needed for proper interpretation of the IR data which will be the subject of a future specialised paper.

"Frequency calculations on the minimised structure of $\text{CH}_2\text{O}@\text{C}_{60}$ gave three translational modes. That corresponding to movement along the long axis of CH_2O is at 233 cm^{-1} , surprisingly good agreement with the observed peak at 231 cm^{-1} (Supplementary Table 10). Calculations also give translational modes in the plane of the CH_2O at 202 cm^{-1} , and perpendicular to that plane at 175 cm^{-1} , compared to the observed single peak at 167 cm^{-1} . DFT neglects the delocalised wave nature of nuclei, treating them as fixed points (Born-Oppenheimer approximation) whereas the observation of ortho and para hydrogen spin isomers due to rotation about the C-O axis indicate that the hydrogen nuclei are delocalised, even at cryogenic temperatures, so the observation of a single peak for translation perpendicular to the C-O axis is reasonable."

3. The word "supermolecular" is used only once in the text, the more accurate term "endohedral" appearing elsewhere. For the sake of uniformity, it should be replaced accordingly.

We used this only in the abstract as it is a much more generally used term and will be recognised by a wider readership than “endohedral” so would prefer to keep.

Reviewer #2 (Remarks to the Author):

- The calculated energies for entry and escape of the small molecules were shown in the main text and Extended Data Table 1. However, the similar data has been already reported in ref. 42 & 44. It is necessary to show that the present data are similar to the reported data or different from them, with citation of the previous data and comments based on comparison with them. In the Extended Data Table 1, please replace “<-” with other description to show chemical equilibrium.

We did not explicitly mention the calculations in ref 42 (on CH₂O@1) in the paper as the omission of correction for basis set superposition error makes the comparison not useful, and in any case the interest is in CH₂O@2. We have added as a note to the supplementary table 6:

“T The activation energy and binding energy (ΔG at 298K) for entry of CH₂O into a model for 1 in which the pyridyl tert-butyl groups are replaced by hydrogens using M06-2X / 6-31G* has been reported¹ as 16.6 kCal/mol (69.5 kJ/mol) and -9.9 kCal/mol (-41.4 kJ/mol) without counterpoise correction for the basis set superposition error² which makes a significant difference (our M06-2X/cc-pVTZ//cc-pVDZ values are 69.6 and -25.0 kJ/mol without correction). Our calculations at the M06-2X/6-31G* level give ΔG for entry and binding of 65.8 and -35.0 kJ/mol (15.7 and -8.4 kCal/mol) respectively without counterpoise correction in good agreement with the published values above, but these become 90.4 and -11.8 kJ/mol (21.6 and -2.8 kCal/mol) with correction illustrating its impact, particularly with small basis sets.

1. Futagoishi, T., Murata, M., Wakamiya, A. & Murata, Y. Encapsulation and Dynamic Behavior of Methanol and Formaldehyde inside Open-Cage C₆₀ Derivatives. *Angew. Chem. Int. Ed.* **56**, 2758-2762 (2017).
2. Boys, S. F. & Bernardi, F. The calculation of small molecular interactions by the differences of separate total energies. Some procedures with reduced errors. *Mol. Phys.* **19**, 553-566 (1970).

Reference 44 reports a value of 4.3 kcal/mol (18.0 kJ/mol) for entry of CO₂ into a model for 1 in which the pyridyl tert-butyl groups are replaced by hydrogen. The only information in the paper or supplementary on how the calculations were carried out is “The structures were optimized at the M06-2X/6-31G* or M06-2X/3-21G levels without any symmetry assumptions”

I would guess that they did not use counterpoise correction for BSSE and they do not say if this is the electronic energy or if frequency calculations have been used to correct for zero point energy, or to correct to ΔH or ΔG at 298K, so any comparison with our calculations would be meaningless.

Given that, and the fact that we are only reporting values for entry into 2 (not 1), we do not think that adding a note is appropriate.

“<-” has been replaced with \rightleftharpoons .

- In the synthetic part in the main text and Figure 1, please make it clear whether the reaction conditions and yields of the products are similar or different from those for the previous examples. It would be helpful if such comparison is clearly provided including the conditions and yields for the synthesis of H₂O@C₆₀, HF@C₆₀, CH₄@C₆₀, Ar@C₆₀, and Kr@C₆₀, to see if the size of HCHO gives some influence or not.

For the Photochemical orifice closure we have clarified the yields for the two species (CH₂O and H₂O filled), which is the strongest evidence for inhibition by CH₂O. We have also mentioned the yields for CH₄, Ar and Kr in the text, although the comparisons of absolute yields are of limited use as we improved the apparatus used since these preparations (use of a central LED light source), for example at the time we did the CH₄@4 synthesis the yield for water filled species was only 25%.

“Allowing for the change in filling factors the overall yields for CH₂O@4 and H₂O@4 can be estimated as 15% and 64% respectively. For comparison previously reported yields over these steps of CH₄@4, Ar@4 and Kr@4 were 13%, 23% and 21%, although an improved procedure using a more powerful light source was used in the current method.

- The filling factors and the chemical yields of the final step of closing for HCHO-encapsulating materials, CO₂-encapsulating material, and labelled materials are rather difficult to understand. It is highly recommended to show the results using chemical scheme with exact numbers of the chemical yields and filling factors. It would be very nice if some comments and data are added to explain the results based on the stabilization/destabilization energies by DFT calculations.

For the final closure of CH₂O@4 to CH₂O@C₆₀ we have added the sentence

“Allowing for the filling factors the yield for CH₂O@C₆₀ can be estimated as 29% (c.f. 52% for the H₂O@C₆₀+C₆₀).”

There were no CO₂ encapsulated materials at this step.

We do not have DFT calculations to examine this process – the suggested mechanism is complex, and the key retro-[4+4] step is orbitally forbidden, so probably occurs via biradicals or an excited state which would require coupled-cluster calculations – not viable for systems of this size.

- The X-ray analysis is not fully completed. It is needed to apply disorder models toward the HCHO moiety, which would sit in different orientations with appropriate disorder factors.

The CH₂O is freely rotating at 100K so X-ray is seeing (the electrons from) a probability distribution of a continuous nuclear wavefunction rather than a mixture of fixed orientations.

During refinement of the crystal structure, attempts were made to describe the CH₂O using various disordered models*, these proved unsatisfactory and it was decided a model with large ellipsoids described the observed electron density distribution best – we were also concerned that presenting a disordered model would be over interpreting the data.

* The CH₂O was modelled as for parts using constrained geometry and thermal parameters. The occupancies refine to approximately 25% in each position and the 4 orientations (figure 1) sit within the broad electron density envelope presented in the paper – we do not believe this model adds any scientific value but does add a lot of additional parameters and constraints to the model.

Figure 1. Electron density represented using a 4-part disorder model.

Another approach tried was to refine the CH₂O split over 2 parts with opposite alignments (i.e. the positions of the C and O switched). The natural tendency of the least squares refinement is to switch these back again and fit both oxygens to the higher concentration of electron density. This is shown in the figure 2 below at various stages of the refinement.

Figure 2. The switched starting model (a) and its refinement (b) & (c).

The most ‘honest’, if unconventional representing is the residual electron density map. We have moved this into the main paper with extended discussion as it also answers a referee 3 comment.

“**Crystal structure of CH₂O@C₆₀.** A crystal of the nickel(II) octaethylporphyrin/benzene solvate of CH₂O@C₆₀ was obtained and subjected to X-ray crystallography. Unconstrained attempts at a structure solution gave an unreasonably short C-O bond. Treating the CH₂O molecule as a rigid fragment with idealised geometry and refining its orientation in space against the observed electron density gave a satisfactory solution (R factor 4.97%) (Fig. 2a). The large thermal ellipsoid on oxygen is one indication of high mobility of the endohedral CH₂O at 100K. To gain greater insight the electron density due to the CH₂O molecule was calculated from the difference between that observed for the entire structure in the X-ray, and that calculated from a model containing all the atoms except the CH₂O. The residual electron density maps (Fig. 2b, c, d) show 2 maxima corresponding to a favoured orientation of the C-O bond axis. DFT calculations on the structure (Supplementary Fig. 27) predict the distances of the oxygen and carbon of the CH₂O molecule from the centroid of the C₆₀ (shown as a green dot in Fig 2a,c) to be 0.86 and 0.34Å respectively confirming the identification of these atoms in Fig. 2a. Fig 2c shows the oxygen atom distributed over a wide arc.”

- Regarding the redox potentials, the authors need careful check on all the results: Although the first reduction of HCHO@C₆₀ is shown to be more negative in Figure 3, however, it is less negative in Extended Data Fig.3, Extended data Table 2, and main text. If the more negative redox potential for HCHO@C₆₀ is correct, the story of main text should be largely modified.

An error was made in the labelling of the DPV curves which has been corrected.

It would be possible to add discussion on the LUMO level obtained by DFT calculations.

We have added a section on DFT calculations to the paper (and supplementary) to answer this and comments from other referees. The relevant part for the LUMO (and HOMO-LUMO gap) is:

“The positions of the HOMO and LUMO of CH₂O@C₆₀ are strongly influenced by the presence of the endohedral molecule (Figure 9). The LUMO energy of CH₂O@C₆₀ is calculated to be 35 meV lower than C₆₀ which compares well to the 30 mV reduction in the first reduction potential found electrochemically. The HOMO is little changed (+3.5 meV) giving a calculated HOMO-LUMO gap for CH₂O@C₆₀ 38 meV smaller than C₆₀ (c.f. 25 meV reduction estimated experimentally from the difference in optical band gaps above). The excitation energy to the 1st excited states of CH₂O@C₆₀ and C₆₀ were calculated using TD-DFT (B3LYP-D3/cc-pVTZ) and the former found to be 23.8 meV smaller corresponding to a 6.9 nm increase wavelength of the longest wavelength absorption in the UV-vis spectra, in excellent agreement with that observed (6 nm change).”

- Although the T₁ value of the ¹H signal for HCHO@C₆₀ is reported to be extremely long such as 30 s, discussion on it looks too short. It would be interesting to compare the data with those for free HCHO dissolved in ODCB, HCHO@2, HCHO@3, HCHO@4, and HCHO@5 under the same temperature and magnetic field, to see if confinement effects are major factor or not.

Additional ¹H T₁ measurements were performed on HCHO@2 and on free HCHO at the same temperature and magnetic field. The other intermediates were not measured due to lack of sample availability. Due to the rapid polymerisation of free HCHO when in solution, the experiment was not possible in ODCB, and tetrahydrofuran (THF) was used as solvent instead. Data and analysis of the T₁ values have been added for these samples in the main manuscript and in the supporting information.

“A similarly long ¹H T₁ relaxation time of 30.8 ± 0.8 s was observed for free monomeric CH₂O in a THF solution (Supplementary Fig 24). In contrast, the formaldehyde ¹H relaxation in the open-cage system CH₂O@2 was observed to be much faster: T₁ = 2.31 ± 0.03 s for CH₂O@2 in ODCB-*d*₄ (Supplementary Fig. 23). The large difference between the ¹H T₁ for the formaldehyde protons in CH₂O@C₆₀ and CH₂O@2 is attributed to the low-symmetry confining environment in the latter case, which partially couples the formaldehyde orientation to that of the open-cage fullerene. Hence the long rotational correlation time of the open fullerene cage leads to a short ¹H T₁ for the endohedral CH₂O in CH₂O@2. This effect is absent for the symmetrical CH₂O@C₆₀ complex, where the rotational motion of the formaldehyde guest is strongly decoupled from that of the fullerene host.”

Reviewer #3 (Remarks to the Author):

1. The authors should comment if the presence of the nickel(II) octaethylporphyrin in the X-ray study has some effect on the orientation of the inner CH₂O molecule.

The paper already included “The residual electron density map when CH₂O is not included (see Extended Data, Figure 2) shows 2 maxima for the carbon and oxygen corresponding to a favoured orientation of the C-O bond axis”.

We have moved the residual electron density figures into the main paper (Fig 2b,c,d) and expanded the discussion of the crystal structure as in the response to referee 2 above..

2. Although provided in the methods section, the electrochemical experimental conditions, namely electrode and solvent used should be included in the main text (or alternatively in the figure caption). In this regard, I wonder if the presence of the formaldehyde carbonyl group is the responsible for this LUMO decrease observation, or simply the proximity of the oxygen atom.

The conditions have been added to the figure caption.

Calculations indicate that the main reason for the change is the distortion of the cage, but the electronic effect of the endohedral molecule also has an effect (but cannot distinguish between the presence of the oxygen and the carbonyl group, although no significant mixing of shell orbitals and CH₂O orbitals is observed in DFT calculations).

3. An in dept NMR study has been carried out with amazing results. In this regard, the study confirms that formaldehyde molecule behaves as in the gas phase. However, the authors could provide an insight to the observation that the ¹³C nuclei of the encapsulated formaldehyde have a strong spin-rotation relaxation mechanism, while the ¹H nuclei do not.

Additional discussion of the spin-rotation mechanism in formaldehyde has been added to the main manuscript. The large difference between the ¹³C and ¹H T₁ relaxation rates is based on the very different magnitudes of the respective spin-rotation tensors. The spin-rotation tensors are taken from experimental data on free formaldehyde molecules, from prior literature now cited in the main text.

4. The spin-isomer conversion of formaldehyde has never been observed experimentally in the condensed

phase. However, in this condensed phase there is not interaction in between formaldehyde molecules. Although I am not an expert in this field, I wonder if it is fair to say condensed phase.

The endofullerene material is in the condensed phase (i.e. solid or liquid). The endohedral formaldehyde behaves similar to the gas phase but the sample itself is a solid. The unusual wording stems for the exotic nature of endofullerenes. No changes have been made.

Reviewers' Comments:

Reviewer #1:

Remarks to the Author:

The authors revised the manuscript to my satisfaction. I recommend its publication as is.

Reviewer #2:

Remarks to the Author:

The reviewer #2 checked the author's responses toward the questions and suggestions. The reviewer found that the authors have added necessary data and description and that some part have been polished well. The reviewer would like the authors to make further comments or responses toward the following minor points:

- line 338; is "(Fig. 9)" wrong? Is it "(Fig. 7)"? And, on Figure 7, it would be helpful if HOMO and LUMO of empty C60 are also presented to understand the differences between those of HCHO@C60 and empty C60, either in the main manuscript or supporting information.

-line 86-88 and 458-466; the reviewer understands that the authors used some specially-prepared setups for the photochemical reaction. It would be better to add detailed information, such as photo of the reaction vessel and light bulbs and precaution needed to run the reaction, which is described only as "A purpose-built reaction vessel" in line 458.

After minor modification, the reviewer believes that this manuscript will be suitable for Nature Communications.

Reviewer #3:

Remarks to the Author:

The authors have nicely addressed the previous concerns/suggestions given from my side, but also by the other reviewers. Therefore, I feel that the quality of the manuscript has significantly been improved and the manuscript in its present form meets the criteria of novelty and quality to be accepted for publication in Nature Communications.

Congratulations to the authors for this nice piece of paper.

Response to Reviewer #2 (Remarks to the Author):

The reviewer #2 checked the author's responses toward the questions and suggestions. The reviewer found that the authors have added necessary data and description and that some part have been polished well. The reviewer would like the authors to make further comments or responses toward the following minor points:

- line 338; is "(Fig. 9)" wrong? Is it "(Fig. 7)"?

Corrected

And, on Figure 7, it would be helpful if HOMO and LUMO of empty C₆₀ are also presented to understand the differences between those of HCHO@C₆₀ and empty C₆₀, either in the main manuscript or supporting information.

This is not useful. C₆₀ has five degenerate HOMOs and 3 degenerate LUMOs, and in any case there is no defined orientation without the endohedral species. The point of the HOMO/LUMO pictures is to show that in CH₂O@C₆₀ they are oriented relative to the CH₂O.

-line 86-88 and 458-466; the reviewer understands that the authors used some specially-prepared setups for the photochemical reaction. It would be better to add detailed information, such as photo of the reaction vessel and light bulbs and precaution needed to run the reaction, which is described only as "A purpose-built reaction vessel" in line 458.

We have added a photograph of the reaction set-up used to the Supplementary information as Figure 29 and referred to it in the methods section when describing the reaction.